# A Novel Blood Proteomic Signature for Prostate Cancer

**DOI:** 10.3390/cancers15041051

**Published:** 2023-02-07

**Authors:** Ammara Muazzam, Matt Spick, Olivier N. F. Cexus, Bethany Geary, Fowz Azhar, Hardev Pandha, Agnieszka Michael, Rachel Reed, Sarah Lennon, Lee A. Gethings, Robert S. Plumb, Anthony D. Whetton, Nophar Geifman, Paul A. Townsend

**Affiliations:** 1Manchester Cancer Research Centre, Division of Cancer Sciences, Faculty of Biology, Medicine and Health, University of Manchester, Manchester M13 9PL, UK; 2Stoller Biomarker Discovery Centre, Division of Cancer Sciences, Faculty of Biology, Medicine and Health, University of Manchester, Manchester M13 9PL, UK; 3The Hospital for Sick Children (SickKids), 555 University Ave, Toronto, ON M5G 1X8, Canada; 4School of Health Sciences, Faculty of Health and Medical Sciences, University of Surrey, Guildford GU2 7XH, UK; 5School of Biosciences, Faculty of Health and Medical Sciences, University of Surrey, Guildford GU2 7XH, UK; 6School of Life Sciences, University of Dundee, Dundee DD1 4HN, UK; 7Salford Royal NHS Foundation Trust, Salford Royal Hospital, Salford M6 8HD, UK; 8Inserm, EHESP, Irset (Institut de Recherche en Santé, Environnement et Travail)—UMR_S 1085, University of Rennes, 35042 Rennes, France; 9Manchester Institute of Biotechnology, Division of Infection, Immunity and Respiratory Medicine, Faculty of Biology, Medicine and Health, University of Manchester, Manchester M13 9PL, UK; 10Murdoch University, Perth, WA 6150, Australia; 11Veterinary Health Innovation Engine, Faculty of Health and Medical Sciences, University of Surrey, Guildford GU2 7XH, UK

**Keywords:** prostate cancer, clinical onset, biomarkers, proteomics, SWATH-MS, complement cascade

## Abstract

**Simple Summary:**

Despite intensive research, effective tools for detection and monitoring of prostate cancer remain to be found. Prostate-specific antigen (PSA), commonly used in prostate cancer assessments, can lead to overdiagnosis and overtreatment of indolent disease. This highlights the need for supporting non-invasive diagnostic, prognostic, and disease stratification biomarkers that could complement PSA in clinical decision-taking via increased sensitivity and specificity. In order to address this need, we uncover novel prostate cancer protein signatures by leveraging a cutting-edge analytical technique to measure proteins in patient samples. This strategy was used as a discovery tool to identify changes in protein levels in the serum of newly diagnosed patients as compared with healthy controls; the feature set was then further validated by reference to a second cohort of patients, achieving a high discriminatory ability. The proteomic maps generated also identified relevant changes in biological functions, notably the complement cascade.

**Abstract:**

Prostate cancer is the most common malignant tumour in men. Improved testing for diagnosis, risk prediction, and response to treatment would improve care. Here, we identified a proteomic signature of prostate cancer in peripheral blood using data-independent acquisition mass spectrometry combined with machine learning. A highly predictive signature was derived, which was associated with relevant pathways, including the coagulation, complement, and clotting cascades, as well as plasma lipoprotein particle remodeling. We further validated the identified biomarkers against a second cohort, identifying a panel of five key markers (GP5, SERPINA5, ECM1, IGHG1, and THBS1) which retained most of the diagnostic power of the overall dataset, achieving an AUC of 0.91. Taken together, this study provides a proteomic signature complementary to PSA for the diagnosis of patients with localised prostate cancer, with the further potential for assessing risk of future development of prostate cancer. Data are available via ProteomeXchange with identifier PXD025484.

## 1. Introduction

Prostate cancer (PCa) is the second most common cancer in the UK, accounting for 13% of all cancer cases [1]. In the USA, it is the third most common type of cancer, occurring in ~19 cases per 100,000 [2]. In the UK, the National Institute for Health and Care Excellence (NICE) guideline (NG131) predicted a 16% rise in the annual cost between 2010 and 2020, from GBP 276.9 million to GBP 320.6 million for the inpatient treatment of PCa [3]. An estimate of annual expenditure to Medicare US, associated with the detection of PCa in elderly men (>70 years old) is USD 1.2 billion/year [4]. Healthcare providers need the means for early diagnosis of the localised, intermediate- to high-risk disease using reliable diagnostic tests that reduce disease burden and increase quality of life (QoL) for elderly men, by lowering associated over-treatment, a significant concern for patients.

Currently, prostate-specific antigen (PSA) is the most common of only a few blood-based diagnostic molecular markers, used alongside digital rectal examination and multiparametric magnetic resonance imaging (mpMRI) in the assessment and diagnosis of PCa. Diagnostic PSA levels are age-stratified; a PSA cut-off of 3 µg/L is used to indicate presence of PCa in men aged 50–59 years [5], while higher PSA values (>3 µg/L) are required to elicit the suspicion of PCa [6] in older men. Nevertheless, elevated levels of PSA are not always age-related and not necessarily indicative of PCa and, therefore, do not offer an accurate diagnostic tool. Elevated PSA can be secondary to non-neoplastic aetiology, such as benign prostate hyperplasia, prostatitis, and/or urinary tract infections [7], thus leading to increased false-positive indications of PCa [8]. This often leads to unnecessary and potentially harmful biopsies; a review by SEER-Medicare (Surveillance, Epidemiology and End Results) reported a 2.65-fold increased risk of hospitalisation within 30 days in approximately 17,000 men having undergone a prostate biopsy [9]. Additionally, there are reported PCa cases with PSA level lower than 4 ng/mL, resulting in false negatives and missed diagnoses [10]. Hence, at an early stage of disease, there is no reliable PSA range that is an explicit signifier for the presence of PCa. Similarly, biopsies can result in false negatives at the early stage of PCa when the tumour is small and the cancer cells are heterogeneously distributed [11,12]. The need to find a highly sensitive and specific biomarker(s) for early screening and PCa diagnosis, potentially complementary to PSA measurement, is essential in improving patient stratification and clinical outcomes.

The need to identify new biomarkers is also essential for the constant monitoring of PCa patients following their diagnosis. While some localised PCa require immediate treatment, such as radical prostatectomy, external beam radiation therapy, or prostate brachytherapy, other patients can be closely monitored over time and placed upon active surveillance; this is reserved for men with low-risk disease, who are eligible for radical treatment but want to avoid the morbidity associated with such therapies until the disease becomes clinically significant. Such decisions are made on the basis of multiple clinical and demographic factors which include PSA, Gleason score (GS), TNM staging, and lower urinary tract symptoms, along with the age and performance status of the patient [13]. Risk to disease progression is more accurately assessed by the D’Amico [14] methodology, which is essential for the management of PCa, therefore, guiding treatment options. While there is an excellent survival rate following treatment of localised PCa, recurrence rates at 7–10 years post-treatment are 20–40% [15]. Altogether, this highlights the need for the identification of new biomarkers from non-invasive monitoring methods to support early screening.

Analysis of the blood proteome using mass spectrometry can offer the optimal biological readout for the identification of new PCa biomarkers [16]. Such approaches are emerging as useful research and investigative tools for biomarker discovery [17] with direct clinical utility [18]. One such method is the Sequential Window Acquisition of All Theoretical Fragment Ion Mass Spectra (SWATH) combined with reverse-phase liquid chromatography used to extract relative fragment ions within the detectable range [19,20,21]. This label-free method allows for a reliable and precise quantification of the proteome across hundreds of samples in combination with novel data extraction approaches [22].

In this study, we identify biomarkers that could improve the diagnosis of PCa by leveraging SWATH-MS as a state-of-the-art analytical platform for discovery proteomics coupled with optimised data analysis tools to investigate the proteomic signatures in the serum of newly diagnosed PCa patients and their age-matched controls. We further assess the relevance of these biomarkers in characterising newly diagnosed PCa patients by addressing the impact of treatments (prostatectomy and radiation therapy) on their relative levels.

## 2. Materials and Methods

### 2.1. Patients

Serum samples were obtained from the SUN Biobank (NHS Ethics REC reference 18/YH/0314). Serum samples were selected from “newly diagnosed” PCa patients (PCa-ND, *n* = 99) and their age-matched healthy controls (HC, *n* = 132) (Table 1). There were 12 participants removed from the study due to incomplete metadata (of which, 11 were PCa-ND and 1 was HC). Control samples (HC) satisfied both normal DRE and PSA levels below 1 ng/mL (<1 ng/mL). The inclusion criteria for PCa-ND patients were an abnormal prostate on digital rectal examination (DRE), symptomatic patients with high prostate-specific antigen (PSA) levels, and abnormal biopsy; or alternatively, a diagnosis made solely on steep rise in PSA associated with urinary symptoms. Patients categorised as Pca-ND were split into two groups: active surveillance (Pca-AS) with no steep progression in their disease, age above 80 years, or life expectancy less than 5 years, while pre-treated patients (PCa-pre) required upfront treatment. PCa-pre comprised patients who underwent two distinct types of treatment to treat cancer: radical radiotherapy (pre-radiotherapy; *n* = 25) and radical prostatectomy (pre-prostatectomy; *n* = 22). Patients having undergone treatment (PCa-post) had been subjected to either prostatectomy or brachytherapy/radiotherapy for (T1- and T2-staged patients), external beam radiotherapy, or prostatectomy, when diagnosed alongside urinary complications (T3 stage).

### 2.2. Blood Collection and Serum Isolation

Peripheral blood was collected into red-capped BD Vacutainer^®^ blood collection tubes (BD Biosciences, San Jose, CA, USA), inverted five times and incubated at room temperature for 30 min before centrifugation at 3000 r.p.m. for 10 min. All samples were processed within 2 h of collection, and the clear fraction (serum) was stored at −80 °C until analysis. All participants provided written consent for the use of data and samples in this study. The study was approved by Yorkshire and the Humber–Leeds East Research Ethics Committee under the reference no. 08/H1306/115+5 and IRAS project ID 3582.

### 2.3. Serum Immunodepletion and Filtration

Samples were prepared according to the Stoller Discovery Centre standard serum proteomic methodology [18]. Immunodepletion of the highly abundant proteins from serum was performed using a Pierce™ Top 12 Abundant Protein Depletion Spin Columns following manufacturer’s instructions (ThermoFisher Scientific, Hemel Hempstead, UK). These 12 proteins were: α1-Acid Glycoprotein, Fibrinogen, α1-Antitrypsin, Haptoglobin, α2-Macroglubulin, IgA, Albumin, IgG, Apolipoprotein A-I, IgM, Apolipoprotein A-II, and Transferrin. Any remaining traces of these depleted proteins by MS were removed from the analysis during first step of data processing. Depleted serum samples were concentrated and purified using Amicon^®^ Ultra-0.5 centrifugal filter devices (Cat #: UFC5003BK, Sigma-Aldrich, Merck KGaA, Darmstadt, Germany). Resulting serum samples were submitted to a buffer exchange using 25 mM ammonium bicarbonate (Sigma-Aldrich, Merck KgaA, Darmstadt, Germany) to decrease salt concentration.

### 2.4. PSA Measurement, Protein Digestion, and Peptide Isolation

PSA levels were assessed using the ADVIA Centaur automated chemiluminescence system (Siemens, Dublin, Ireland) [23]. Bicinchoninic acid assay (BCA assay, Thermo Fisher Scientific, Hemel Hempstead, UK) was used to quantify the total protein content of concentrated serum samples. The total protein content was then normalised to 50 μg per 96 μL of sample using 25 mM ammonium bicarbonate. The amount of 5 mM dithiothreitol (GE Healthcare Life Sciences, Parramatta, Australia) and 1% (*w*/*v*) sodium deoxycholate (Sigma-Aldrich, Merck KgaA, Darmstadt, Germany) was used to reduce proteins at 60 °C for 30 min. Capping of reduced cysteine residues was achieved using 10 mM iodoacetamide (Sigma-Aldrich, Merck KgaA, Darmstadt, Germany) at room temperature for 30 min in the dark. Trypsin (Promega, Southampton, UK) was added for the digestion of alkylated proteins in an enzyme:protein ratio of 1:50 at 37 °C for 16 h. Sodium deoxycholate was removed from serum samples using a 0.5% (*w*/*v*) solution of formic acid (Fisher Scientific, Thermo Fisher Scientific, Waltham, MA, USA). The peptide-containing supernatants were dried-down using a MiVac Quattro Concentrator (Genevac™, Thermo Fisher Scientific, Waltham, MA, USA) for 2–3 h and dried pellet stored at −80 °C until further analysis.

### 2.5. SWATH Analysis

Dried peptide samples were reconstituted in a mixture of loading buffer (2% acetonitrile/0.1% formic acid), pepcalmix (10 fmol/uL, SCIEX, Warrington, UK), and index retention time (iRT) peptides (2X) (Biognosys, Schlieren, Switzerland) for analysis using the sequential window acquisition of all theoretical fragment-ion spectra (SWATH) mass spectrometry platform [23]. Acquisition of samples was performed on a 6600 TripleTOF mass-spectrometer (hereinafter called MS) (SCIEX, Warrington, UK) coupled with Micro HPLC system (SCIEX, Warrington, UK) with an analytical column (YMC-Triart C18 12 nm, 3 µm, 0.3 mm I.D. × 150 mm, 1/32″ column; YMC Europe GmbH TA12S03-15HORU) and trap Column (YMC-Triart C18 12 nm, S-3 µm, 5 mm × 0.5 mm I.D., 1/32″; YMC Europe GmbH TA12S03-E5JORU). The sample pickup volume was set to 8 μL at a flow rate of 5 μL/min. The 100 variable window method was used along with MS parameters to acquire the data: accumulation time 0.249988 s, *m/z* range 400–1250, duration 119.987 min, cycles 2572, delay time 0 s, and cycle time 2.7991 s [18]. The human leukemia K562 cell line was used to obtain a cell lysate protein extract digest used as quality control (QC) alongside a control serum (Seralab, NC3Rs, London, UK). Control samples were loaded every 4 samples to ensure consistent instrument performance across runs and to highlight issues with sample processing. Raw data files (wiff) were searched against a purpose-built PCa serum library [24] (acquired through data-dependent acquisition MS) using openSWATH (version 2.0.0) [25]. PyProphet (version 0.18.3) [26] was used to score the peptide matches, and results were aligned using the Msproteomicstools feature alignment [27].

### 2.6. Processing of MS Proteomic Data

SWATH2Stats package (Bioconductor packages release 3.11) [28] was used to annotate the feature alignment file and filter the data at an m-score cutoff of 0.01. All SWATH-MS runs with a transition level FDR (fragment ion FDR cutoff) greater than 0.03 were excluded from the analysis. Data were converted into a format readable by Msstats package (Bioconductor packages release 3.11) [29]. Data were processed by choosing the ‘top3’ feature subset and normalization of the resultant information by “equalizeMedians”. SummaryMethod was set to “TMP”, and cutoffCensored was set to “minFeature”, with no imputation of missing information. In the acquired protein quants, all the “NA” were replaced by “0”; no missing value imputation steps were used [30]. Non-parametric tests (Wilcoxon rank-sum test) were performed using MetaboAnalyst (4.0) [31] to estimate significance among binary variables. One-way ANOVA and post-hoc testing were performed using MetaboAnalyst (4.0) [31] to determine the statistical significance among multiple (more than 2) conditions. SWATH2Stats and SWATH-MS analysis were performed in Rstudio (Version 4.0.2).

### 2.7. Biomarker Analysis and Development and Evaluation of Classification and Regression Models

Statistical analysis was conducted in R. Fold change (FC) analysis compares the absolute value of change between two group means. Fold changes were calculated with the direction of comparison set to PCa—HC using the log-transformed data. The plotnine library in Python was used to visualise the results. Statistical analysis was carried out by unpaired non-parametric Mann–Whitney test, and significance was determined by *p*-values. Classification analysis was conducted in Python using the scikit-learn library (Version 1.02) [32]. A random forest (RF) model was used for feature selection from the initial protein dataset, splitting the data in the ratio of 67:33 (training/testing using stratification), setting ntree to 1000, maximum leaf nodes to 16, and random seed to 1. Recursive feature elimination with cross-validation was used to select the individual features from the training set (maximising accuracy). Classifier model construction was performed using logistic regression. All performance metrics were calculated using the test set only to avoid data leakage. The performance of the classification model on the test set was depicted using a classification matrix and the AUC (area under the receiver operating characteristic (ROC) curve of the discovery model); both were calculated with the Yellowbrick library (Version 1.5) [33]. The data were also visualised by principal component analysis (Python library, Yellowbrick; Version 1.5). Multivariate regression analysis against PSA scores was also conducted in Python using the scikit-learn library. RF models were used to assess both the initial protein data set and the reduced feature set with the same training/testing approach and random seed as described above. Regression analyses were assessed by measuring R^2^ on both the training set and the testing set and by visual inspection of residuals plots.

### 2.8. Functional Annotation and Pathway Analysis

Pathway analysis using the various protein signatures was performed using ClueGO (Version 2.5.7), a plug-in application in Cytoscape (Version 3.8.0) [34,35]. The following databases were searched against: GO: Biological Process (08.05.2020); GO: Molecular Functions (08.05.2020); GO: Immune System Process (08.05.2020); KEGG (08.05.2020); Reactome: Pathways (08.05.2020); and WikiPathways (08.05.2020). Only pathways with *p*-value < 0.05 and with a minimum of 3 proteins per pathway were considered. Pathway enrichment/depletion analysis was performed in a two-sided hypergeometric test and using Bonferroni step-down correction. The specific proteins (from the biomarker list) characterising each pathway/cluster were highlighted in the generated graph.

## 3. Results

### 3.1. Serum Proteome Reveals a Signature of Newly Diagnosed Prostate Cancer Patients

To identify a specific, novel proteomic signature for clinical onset in PCa patients, we obtained serum samples from the SUN prostate cancer biobank (Table 1). From this cohort (consisting of patients at multiple stages of PCa), we analysed 88 PCa patients for whom serum samples were available at time of diagnosis; who had no prior history of PCa, other cancers, or debilitating comorbidities; and for whom complete metadata were available. These newly-diagnosed PCa patients (PCa-ND) were split into two groups on the basis of the subsequent choice of treatment: patients put on active surveillance (PCa-AS, *n* = 41) and those requiring immediate treatment at time of diagnosis (PCa-pre, *n* = 47). Particular attention was paid to limiting confounding factors between the two groups with regard to PSA levels, age, Gleason score, or tumour stages (Table 1). Healthy aged-matched controls (HC) (*n* = 131) showed significant lower levels of PSA below 1 ng/mL (0.81 ± 0.5 ng/mL versus 10.46 ± 17.94 ng/mL, *p*-value = <0.0001) and were devoid of any cancer or other known comorbidities. A total of 336 proteins were identified using our proteomic workflow (Appendix A). Of these, 12 proteins were found to discriminate between HC and PCa-ND as selected by random forest using RFECV, among which 6 were significantly (*p*-value < 0.05) up-regulated in PCa-ND (THBS1, C1QA, C1QC, CFHR2, IGHG1, and IGKV1-39); 3 in HC (SERPINA5, APOC2, and APOE); and 3 proteins showed statistically non-significant differential values between cases and controls (ECM1, GP5, and GP1BA; *p*V > 0.05). These features are shown in Figure 1a and also in Appendix A.

The ability of the biomarker signatures to differentiate between the PCa and HC groups was visualised by principal component analysis; the 12 putative biomarkers allowed for the distinct separation between HC and PCa, with limited overlap (Figure 1b). A logistic regression model based on the reduced feature set described above generated a strongly predictive signature differentiating between HC and PCa-ND (AUC = 0.93, Figure 1c). Sensitivity for the diagnostic model derived from the test set confusion matrix (Figure 1d) was 0.77 (95% confidence interval 0.58–0.90), specificity was = 0.93 (0.81–0.99), and overall diagnostic accuracy was 0.86 (0.77–0.93). Analysis of the ranked importance of each putative biomarker in the model showed thrombospondin-1 (THBS1) as the most important biomarker.

The addition of PSA level values (as measured by ELISA) further increased the predictive ability of the 12-protein signature (AUC = 1.00). This was an expected outcome in our cohort, where PSA was highly discriminatory between cases and control (as the controls were selected based on PSA levels below 1 ng/mL). A multivariate LASSO regression analysis of the relationship between proteomic signature and PSA score, however, did not find that the proteome was strongly predictive of the PSA score (test R^2^ of 0.145, Appendix A).

The separation between HCs and PCa-AS and PCa-Pre was also investigated; no meaningful separation was achieved. Isolating PCa-AS and PCa-Pre and conducting supervised separation by logistic regression using the same methodology as for the separation of HCs and the overall PCVa-ND cohort showed that the proteome was not overtly different between the two, thus yielding no separation by PCA and AUC of 0.46 (Figure 1e,f).

### 3.2. Validation of Proteomic Biomarkers Using an External, Independent Cohort

As a validation step in our selection of potential biomarkers, we identified and used an appropriate previously published independent dataset [21]. In this multi-cancer SWATH-MS study, serum samples were used to identify proteomic markers in early-stage PCa patients; these were sufficiently similar to our samples, thus enabling a comparative analysis. We cross-referenced our initial list of discriminatory proteins against those identified in Sajic et al., selecting only those identified in both studies and where the direction of expression was the same. Given that methodologies and processing/instrumentation were not identical, overlap between the two studies would necessarily be limited; thus, this was a conservative approach to biomarker identification. Of the 12 proteins from the discovery set, 5 were identified with the same direction of expression within the validation cohort: these were GP5, SERPINA5, ECM1, IGHG1, and THBS1, with the latter again having the most discriminatory power. The same modelling and assessment steps conducted for the initial discovery set were repeated and resulted in similar AUC (0.91) but slightly reduced sensitivity (0.66 with 95% confidence interval 0.46–0.82), specificity (0.84 with 95% confidence interval 0.70–0.93), and overall diagnostic accuracy (0.77 with 95% confidence interval 0.65–0.86), as compared with the discovery set. These results are illustrated in Figure 2a–c.

### 3.3. A Central Role for Complement and Coagulation Cascade in Newly Diagnosed Prostate Cancer

To further investigate the biological significance of the biomarker signature identified, a functional enrichment analysis was carried out. The panel of proteins was examined with several functional libraries, and only statistically relevant pathways with a minimum of three common proteins were further investigated (Figure 3). A total of four functional clusters were identified, which included the complement and coagulation cascades, regulation of blood coagulation, complement activation, and regulation of respiratory burst. The detailed functions associated with our proteomic signature further included synapse pruning, microglia pathogen phagocytosis pathway, and oxidative damage. The detailed list of pathways, ontological sources, and FDR-corrected *p*-values is set out in Appendix A.

Of the 12 proteins within our discovery biomarker panel, 4 (THBS1, C1QA, C1QC, and SERPINA5) were associated with several functional clusters (Figure 3), all linked to various pathways related to the coagulation, complement, and clotting cascades, as well as to ligand binding/uptake and to plasma lipoprotein particle remodelling.

### 3.4. Treatment-Related Changes in the Proteomic Signature of Newly Diagnosed Prostate Cancer

We next examined whether the biomarker signature characterising PCa at time of diagnosis was altered in patients who have undergone active treatment. To this end, we examined the levels of the protein biomarkers in PCa patients who had undergone specific Pca treatment, such as radical prostatectomy (*n* = 12) and radiotherapy (*n* = 13), following diagnosis (Table 1). These treated PCa patients presented with significantly lower levels of PSA (0.33 ± 0.62 ng/mL; 3.50 ± 4.56 ng/mL, respectively) compared with their respective baseline pre-treatment samples (pre-prostatectomy 8.63 ± 5.54 ng/mL—*p*V < 0.0001; pre-radiotherapy 11.91 ± 24.68 ng/mL—*p*V = 0.0003), while these levels were higher than basal levels characterising HC (0.91 ± 0.5 ng/mL). Re-evaluation of the expression levels of the core putative biomarkers in PCa patients following radiotherapy treatment identified no reversion in panel proteins, and the linear regression model used to distinguish HC and PCa-ND showed no separation capability when applied to PCa-ND versus PCa-Post participants. No visual separation was obtained by PCA (see Appendix A), with the model classifying both PCa-ND and PCa-Post samples as PCa-ND, i.e., sensitivity of 0.0 with regard to PCa-Post.

## 4. Discussion

The discovery of PSA and widespread usage in the 1990s enhanced the detection and treatment of prostate cancer [36]. Additionally, PSA levels are used to stratify individuals prior to intervention and to monitor their response to local or systemic treatment. Nevertheless, PSA is not specific to PCa: elevated levels of PSA can also be measured in cancer-free individuals with either/both enlarged prostate or prostate infections. Furthermore, within PCa, the range of PSA levels and its association with cancer severity and stage can vary greatly between individuals. This can result in unnecessary biopsies, preventable morbidity, and overtreatment of cancers that are not clinically significant [37].

In recent years, a number of novel biomarkers that appear to outperform PSA have been suggested, with the potential for these to be used to better detect PCa and distinguish between individuals who will benefit from active treatment from those who will not, as well as aid in the improved outcome of surgery [38,39]. These indicators hold the promise of significantly reducing unnecessary biopsies and operations. However, it is crucial to keep in mind that these novel biomarkers have their own limitations and that results from these studies should be interpreted cautiously and in conjunction with current clinical data [40]. The majority of leading cancer guidelines classify these biomarkers as experimental, with many still pending FDA approval. Among them are 4Kscore [41], MiPS [42], Stockholm-3 [43], Intelliscore [44], MdxHealth [45], ProMark [46], Oncotype Dx [47], Decipher [48], PSCA [49], and Proclarix [50]. Due to the undetermined performance of these biomarkers in real-world settings, cancer guidelines do not currently encourage widespread use of these biomarkers. Thus, more research is needed to determine their diagnostic and therapeutic utility [51] in heterogenous populations, and there is still a requirement for further investigation, as undertaken in the current study. It is noteworthy that to date, none of the above clinical tests has yet been demonstrated to improve patient survival.

In this study, SWATH-MS was used to examine the proteome profiles found in serum samples of newly diagnosed PCa patients and their age-matched healthy controls. A 12-protein marker profile showing very good classification performance (AUC of 0.93 and test set diagnostic accuracy of 0.86) was developed (Figure 1). This outperformed previously published assays, such as PSA (AUC = 0.52), PSA density (AUC = 0.70), %fPSA (AUC = 0.75), PHI—Prostate Health Index (AUC = 0.76), PHI density (AUC = 0.84), and the 4K score (AUC = 0.81) [51]. The 12-protein signature performed better than reported for Proclarix (THBS1, CTSD), a new-generation serum PCa marker which demonstrated an AUC of 0.83 when used alone and 0.85 when combined with %fPSA [51]. THBS1 and complement factor, in combination with other protein markers, were initially demonstrated as blood biomarker for prostate cancer by Cima et al., 2011, to accurately predict 78% of patients with aberrant or normal PTEN status with an AUC of 0.82 [52].

Reproducibility of any biomarker signature and performance is critical. The reproducibility of our 12-protein signature was assessed with reference to an independent cohort, retaining only those proteins who showed the same direction of expression in both datasets. Given that the validation study was undertaken using different protocols and on different instruments, the low level of overlap in identified proteins is not surprising and, so, represents a conservative and discriminatory approach to biomarker identification. Nonetheless, the panel of five proteins still resulted in an AUC of 0.91. It should be noted that whilst AUC is a standard metric, where classifier curves cross, AUC can be inflated; thus, test-set diagnostic accuracy, sensitivity, and specificity are more helpful metrics. [53] The five-panel test set diagnostic accuracy was 0.77, still retaining the large part of the diagnostic power of the larger dataset of 12 features.

Our results further demonstrate the utility and applicability of the SWATH-MS technology as an analytical tool for biomarker discovery. Thus, when compared with healthy controls, the five-protein signature is more tractable than previously developed assays (discussed above) in terms of discriminating localised and locally progressed illness. While PSA was not discovered or quantified directly using SWATH-MS, we compared our 12-protein signature’s performance with PSA levels established by ELISA in these samples. In our discovery SUN cohort, PSA alone demonstrated an AUC of >0.9. However, in this cohort, PSA likely has inflated performance since the cancer cases included in our cohort were confirmed via biopsies and a significant rise in PSA, and the HC samples in this study were chosen to have a PSA level of less than 1 ng/mL. This approach for selecting patients removes the likelihood of false positives caused by PSA alone (increasing its overall estimated accuracy). Interestingly, the relationship between PSA and the circulating protein signature developed here was not strong (test R^2^ of 0.145), and whilst PSA in the post-treatment cohort reverted towards HC levels, the protein signature did not. This is suggestive that the protein markers identified provide an orthogonal, rather than a correlated, test to that provided by PSA. This represents a potentially promising result given the goal of developing a complementary test to PSA.

Whilst reliance on PSA in clinical practice has some shortcomings, elevated PSA level acts as a potent factor to separate patients who need immediate treatment from patients who remain on “watchful wait” [54]. Our panel of biomarkers was not able to demonstrate a significant difference in expression between patients who were put under active surveillance and those who were to undergo other active treatments (Figure 1), although interestingly, neither did PSA in this cohort. Nevertheless, we are now in a position to consider the use of these markers for early detection of PCa using prospective cohorts.

The critical role of the innate and adaptive immune system in the control and surveillance of cancer has long been understood, where the complement system is the key inter-player between the two [55]. Complement activation is controlled by binding of C1q to the antigen–antibody complex, which subsequently causes the pathway to progress [56]. This results in the production of C3a and C5a, which are powerful chemo-attractants for macrophages and lymphocytes to modulate tissue damage through production of growth factors, cytokines, and radical oxygen species [57,58,59]. Cancer cells, though recognised by the complement system, guard themselves from complement-mediated lysis by undergoing some structural modification [60]. This, along with supply of complement regulatory proteins, facilitates the cancer growth [61]. CFHR family proteins also play a vital role in inactivation and degradation of complement complexes [62]. There is mounting evidence indicating the upregulation of complement cascade components in the tumour microenvironment and its linkage to tumour growth and metastasis [63,64]. This is in agreement with our own results, where upregulation of C1QA, C1QC, and CFHR2 in newly diagnosed PCa patients is suggestive of the involvement of tumour inflammatory microenvironment, which may act as a mediator of tumour progression and angiogenesis (Figure 3).

The tumour microenvironment itself offers support for tumour survival [65]. As mentioned, among the immune cells recruited to the site of tumour are tumour-associated macrophages (TAM), which are reported at all stages of cancer development and assist in neuroendocrine (NE) differentiation, a hallmark of PCa [66]. Clinical observations have also advised that NE differentiation of PCa correlates with disease development and poor prognosis [67]. Our functional enrichment analysis demonstrated the involvement of highly expressed PCa-ND markers in complement and coagulation cascade, macrophage differentiation, microglia pathogen pathogenesis, and synapse pruning. This was contributed mainly by member of the C1q family, CFHRs family, and PF4. Secretion of platelet chemokine PF4 from activated platelets in atherosclerosis models promotes further macrophage differentiation [68]. Recently, its distinctiveness as an angiogenic marker in castration-resistant PCa patients has also been demonstrated [69,70]. While the central role of macrophages in PCa progression and its likelihood to metastasize to bone is known [71], their association with the neuroendocrine nature of PCa is important as macrophages (and more specifically microglial cells) are responsible in the promotion and modelling of neurogenesis and axogenesis. This signature importantly captures an interplay among the tumour microenvironment, development of nerve fibres, and macrophage functions, which has been shown to be critical for the activation of angiogenesis and promotion of tumorigenesis of PCa [72,73]. It helps in monitoring disease advancement and provides a therapeutic target to diminish the associated complications.

This study has further identified an increase in the expression of THBS1 (Figure 1a). Though THBS1 is recognised as an anti-angiogenic molecule, its role in angiogenesis mediation by stimulating endothelial migration is also well established [74]. Its upregulation has been reported in various cancer types, including prostate and gastric carcinoma [21,73,74], and it was also reported to upregulate MMP9 in breast cancer cells [75]. This is in agreement with our findings, where increased expression of MMP9 in PCa patients with elevated THBS1 and PSA is evident (Figure 1b). In PCa cell lines, LNCaP transfected with MDM2 showed high expression of THBS1 and MMP9, therefore suggesting MDM2 as an upstream regulator of THBS1 and MMP9 expression that tilts the balance towards pro-angiogenic mechanisms [76].

SERPINA5 is an important component of the SERPIN family known as a putative tumour suppressor gene [77]. It inhibits the activation of PSA and kallikrein, and it boosts sperm motility and fertilization in many studies [78]. The decreased expression of SERPINA5 seen in our discovery cohort and elevated PSA levels in newly diagnosed PCa patients compared with healthy individuals fits well with current research. This may also explain reduced fertility in these patients [79]. Further, expression of SERPINA5 was not reverted to normal levels following surgery and exposure to radiation. Similarly, a decrease in expression of SERPINA6, another member of the SERPIN family, was found in newly diagnosed PCa patients compared with baseline healthy individuals, and interestingly, it remained decreased in post-treatment individuals compared with healthy individuals. Its relationship in the development of PCa has yet to be established.

We would highlight limitations in this work. Putative or candidate biomarkers were validated in a second independent cohort. Nonetheless, a larger-scale, targeted, and fully quantitative analysis would be required for definitive validation, alongside biomarker analytical validation against standards for definitive identifications. A larger study would also have the potential to investigate a biomarker panel’s specificity with regard to comorbidities, such as benign prostatic hyperplasia or indeed other diseases, and whether other pathways or markers (such as those associated with autophagy [80]) might be more specific to PCa. Furthermore, the selection of participants with pre-specified PSA levels may have provided a less challenging cohort than a random selection. Notwithstanding these limitations, data-independent acquisition proteomics has the potential to become a clinical tool with decreased run times and increased throughput becoming available [81], with the ultimate goal of providing early screening via multi-measurement assays. Alternatively, selected reaction monitoring (SRM) mass spectrometry or affinity-based (immunoassay) approaches can take this work forward. We will be developing an SRM-based approach compatible with usage in clinical biochemistry laboratories, thus improving sample throughput, precision, and intra-lab applications, while reducing run time and cost. In this respect, moving to other protein assay platforms, such as immunoassays, may not be required. This has the benefit of moving to a more specific detection platform with a wider dynamic range, as offered by mass spectrometry.

## 5. Conclusions

Our study describes the novel use of the SWATH-MS analytical platform, which provides an in-depth, hypothesis-free analysis of the proteome in prostate cancer patients. This has enabled the measurement of proteomics, coupled with optimised data analysis tools, to identify proteomic signatures in the serum of newly diagnosed PCa patients. We have demonstrated and validated a core panel of five protein biomarkers that differentiates between newly diagnosed PCa and healthy controls, highlighting the utility of such biomarker signatures and potentially complementing PSA as a diagnostic tool. Earlier and specific diagnosis would enable improved surveillance and individually tailored treatment options, thereby reducing morbidity associated with radical treatment. Whilst further validation work is required, this work demonstrates the potential for these putative biomarkers for PCa to deliver positive health impacts.

## Figures and Tables

**Figure 1 cancers-15-01051-f001:**
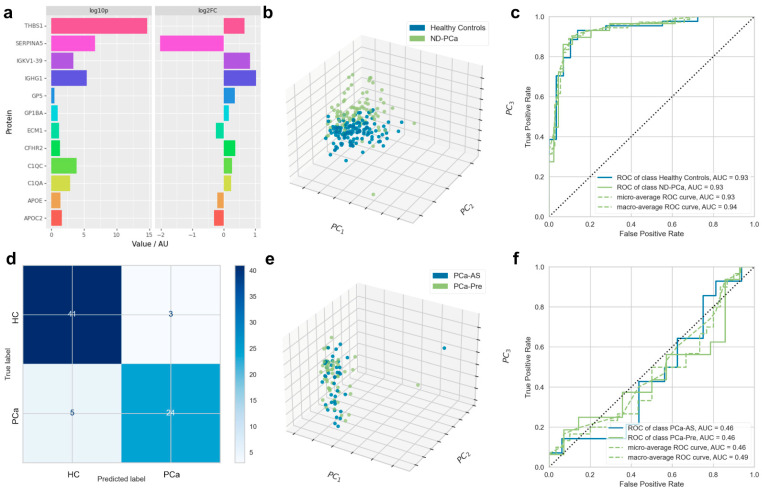
Panel of twelve serum protein biomarkers discriminating newly diagnosed prostate cancer patients and healthy controls. Biomarkers segregating healthy controls (HC) from newly diagnosed prostate cancer patients (PCa-ND) (composed of individuals put on active surveillance or requiring immediate treatment at time of diagnosis) were characterised by random forest analysis; with classification models constructed by logistic regression. (**a**) Significance (−log_10_ (*p*V)) and expression fold change (log_2_ (FC)) of biomarkers differentiating between HC and PCa-ND. (**b**) Principal component analysis (PCA) showing the degree of separation between HC and PCa-ND. The PCA axes show the first, second, and third most important directions in the reduced space along which the samples show the largest variation. (**c**) AUC for test set only showing individual ROC curves for HC and ND-PCa participant classification. (**d**) Confusion matrix for linear regression model applied to PCa versus HC participants. (**e**) The degree of separation between PCa-Pre and PCa-AS. (**f**) AUC for test set only showing individual ROC curves for PCa-Pre and PCa-AS participant classification.

**Figure 2 cancers-15-01051-f002:**
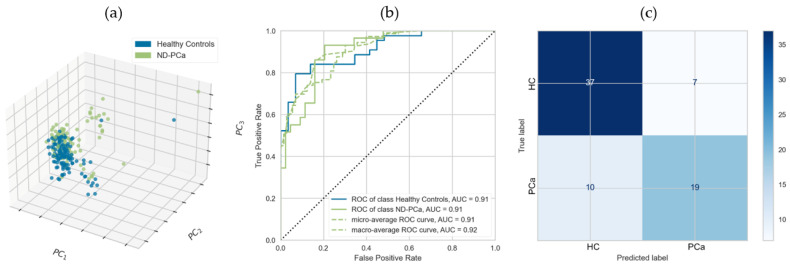
Panel of 5 serum protein biomarkers discriminating newly diagnosed prostate cancer patients and healthy controls. Biomarkers segregating healthy controls (HC) from newly diagnosed prostate cancer patients (PCa-ND) (composed of individuals put on active surveillance or requiring immediate treatment at time of diagnosis) using GP5, SERPINA5, ECM1, IGHG1, and THBS1; with classification models constructed by logistic regression. (**a**) Three-component PCA illustrating separation of Healthy Controls and ND-PCa participants. (**b**) AUC for test set only showing individual ROC curves for HC and ND-PCa participant classification. (**c**) Confusion matrix for linear regression model applied to PCa versus HC participants.

**Figure 3 cancers-15-01051-f003:**
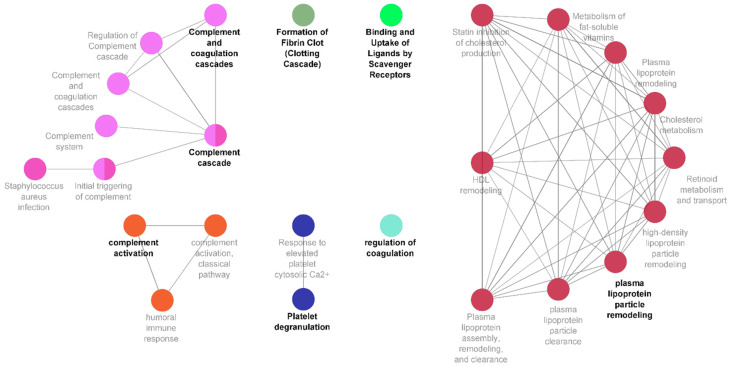
Complement and coagulation cascades characterise newly diagnosed prostate cancer patients alongside regulation of coagulation, clotting cascade, ligand binding and uptake, and plasma lipoprotein particle remodelling. Network representation of specific functional clusters created using the ClueGO application within Cytoscape, derived from those proteins showing statistically significant differences between the PCa and HC classes. Identified proteins associated to these pathways are detailed in Appendix A.

**Table 1 cancers-15-01051-t001:** Clinical profile of patients enrolled and analysed in the study. Patients were segregated into newly diagnosed (PCa-ND) and post-treatment (PCa-post) PCa patients. PCa-ND were composed of individuals put on active surveillance (PCa-AS) or referred for further treatment (PCa-pre) at time of diagnosis. PSA: prostate-specific antigen, HGPIN: high-grade prostatic intraepithelial neoplasia, NA: not applicable. Where specified, ± determines standard deviation, and n represents the number of individuals in each category.

Group	Age (Years)	PSA (ng/mL)	Gleason Score	Tumour Stage
Newly Diagnosed Prostate Cancer Patients (PCa-ND)				
Active Surveillance (PCa-AS) (*n* = 41)	68 (±8)	9.7 (±9.6)	HGPIN (*n* = 1) 3 + 3 (*n* = 33) 3 + 4 (*n* = 1) NA (*n* = 6)	T1–T3 (no nodal spread and no metastasis)
Pre-treatment (PCa-pre) (*n* = 47)	64 (±6)	8.1 (±5.1)	2 + 2 (*n* = 1) 3 + 3 (*n* = 32) 3 + 4 (*n* = 12) NA (*n* = 2)	T1–T3 (no nodal spread and no metastasis)
Post Treatment (PCa-post)				
Post prostatectomy (*n* = 12)	63 (±6)	8.6 (±5.5)	N/A	N/A
Post radiotherapy (*n* = 13)	63 (±6)	11.9 (±24.7)	N/A	N/A
Healthy Controls (HC)				
Healthy Control (*n* = 131)	66 (±10)	0.8 (±0.6)	N/A	NA

## Data Availability

All data needed to evaluate the conclusions in the paper are present in the paper and/or the Appendix A. Raw data files are available via ProteomeXchange with identifier PXD025484 and can be accessed through ProteomeXchange using reviewer account details: Username: reviewer_pxd025484@ebi.ac.uk and Password: W0lWREBN.

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
