# Peer review of "A Novel Blood Proteomic Signature for Prostate Cancer"

_cancers, 2023, doi:10.3390/cancers15041051_

Round 1
Reviewer 1 Report
Muazzam et al reports an interesting study using a proteomic approach, i.e. SWATH-MS, to identify potential proteins biomarkers for an earlier diagnosis of patients with prostate cancer.
The paper in my opinion is well written, the experiments were well planned and conducted, the authors maked a great effort to analyse the list of proteins that they obtained with the SWATH-MS analysis and to identify putative biomarkers linked to specific pathways. It is interesting the possible clinical applicability of these laboratory tests.
I have only a minor comment, the authors have written that they use the ELISA assay (line 289, 421) to measure the PSA level, but no data are available, and in the section of “material and methods” the ELISA test has not been described.
Moreover, I would like to make a little suggestion, the authors could performed some specific biochemical experiments to validate the identified biomarkers.
In conclusion, the present manuscript merits to be published
Author Response
We thank the reviewers for their time and for the helpful comments. We provide point-by-point responses below.
REVIEWER 1
Muazzam et al reports an interesting study using a proteomic approach, i.e. SWATH-MS, to identify potential proteins biomarkers for an earlier diagnosis of patients with prostate cancer.
The paper in my opinion is well written, the experiments were well planned and conducted, the authors maked a great effort to analyse the list of proteins that they obtained with the SWATH-MS analysis and to identify putative biomarkers linked to specific pathways. It is interesting the possible clinical applicability of these laboratory tests.
I have only a minor comment, the authors have written that they use the ELISA assay (line 289, 421) to measure the PSA level, but no data are available, and in the section of “material and methods” the ELISA test has not been described.
We thank the reviewer for highlighting this point: we have added the technical details of the clinical test to the manuscript (Siemens’ ADVIA Centaur system) at line 163 and PSA levels are reported in Table 1.
Moreover, I would like to make a little suggestion, the authors could performed some specific biochemical experiments to validate the identified biomarkers.
We agree that this is an important part of the validation process. An analytical validation (with referenced retention times by liquid chromatography and fragmentation patterns by mass spectrometry) to fully verify the biomarkers against standards was beyond the scope and budget of this work. We have added a comment to the Discussion to highlight that such an analysis would be part of a validation study (lines 506-512).
Reviewer 2 Report
Dear authors, the paper is very interesting, hoping to see these putative newly discovered biomarkers for PCa to deliver positive health impacts, although I agree with you a larger-scale analysis would be required for definitive validation.
Minor point: in line 385 are cited biomarkers recently discovered but the last one published in 2022 was omitted (Ljubic et al 2022).
Author Response
We thank the reviewers for their time and for the helpful comments. We provide point-by-point responses below.
REVIEWER 2
Dear authors, the paper is very interesting, hoping to see these putative newly discovered biomarkers for PCa to deliver positive health impacts, although I agree with you a larger-scale analysis would be required for definitive validation.
Minor point: in line 385 are cited biomarkers recently discovered but the last one published in 2022 was omitted (Ljubic et al 2022).
We thank the reviewer for highlighting this helpful reference and have added it to the manuscript (line 386).
Reviewer 3 Report
The study aim is worthy of interest. Moreover, it focused on the need to identify new biomarkers for PCa. The authors demonstrate the utility and applicability of the SWATH-MS technology as an analytical tool for biomarker discovery. However, a deep makeover of the whole manuscript is required.
- PSA levels and age stratification could be may be incomplete and inaccurate, I recommend removing them from the introduction.
- Please restructure the introduction by shortening and eliminating data and comments from other studies that even if valuable should be included in the discussion. Indeed, some concepts are retracted in the discussion and this makes the whole text redundant and difficult to read.
- The need for new biomarkers for PCa is the central core of the whole paper. For this scope, you should include also a new application for new biomarkers. Moreover, recent findings demonstrated how a novel urine biomarker could predict radically after prostatectomy. Moreover, the measurement of 8-OHdG and of 8-Iso-PGF2αin in urine before and after surgery as a technique to help predict radicality (and perhaps local recurrence) following surgery (DOI: 10.3390/jcm11206102). I strongly believe this should be included in your paper.
- The functioning of autophagy is highly complex, and it interacts with several biological processes. This plays a key role in tumor aggressiveness which you have extensively covered in your manuscript. for this reason, include a consideration regarding the processes of apoptosis by including this interesting novel study on the topic (DOI: 10.3390/ijms23073826)
- Check typos
Author Response
We thank the reviewers for their time and for the helpful comments. We provide point-by-point responses below.
REVIEWER 3
The study aim is worthy of interest. Moreover, it focused on the need to identify new biomarkers for PCa. The authors demonstrate the utility and applicability of the SWATH-MS technology as an analytical tool for biomarker discovery. However, a deep makeover of the whole manuscript is required.
- PSA levels and age stratification could be may be incomplete and inaccurate, I recommend removing them from the introduction.
Whilst we agree with the reviewer that age stratification and PSA levels have issues, they are widely reported indicators with regard to PCa. We have modified the text to reflect the reviewers concerns (line 65); we have retained the data in Table 1 to meet journal standards.
- Please restructure the introduction by shortening and eliminating data and comments from other studies that even if valuable should be included in the discussion. Indeed, some concepts are retracted in the discussion and this makes the whole text redundant and difficult to read.
We have reviewed the Introduction for brevity and clarity (e.g. removing lines 76-78, overlapping with lines 88-91) and hope that this meets the reviewer’s concerns.
- The need for new biomarkers for PCa is the central core of the whole paper. For this scope, you should include also a new application for new biomarkers. Moreover, recent findings demonstrated how a novel urine biomarker could predict radically after prostatectomy. Moreover, the measurement of 8-OHdG and of 8-Iso-PGF2αin in urine before and after surgery as a technique to help predict radicality (and perhaps local recurrence) following surgery (DOI: 10.3390/jcm11206102). I strongly believe this should be included in your paper.
We thank the reviewer for highlighting this helpful reference and have added it to the manuscript (line 386). With regard to new applications, we have added text at lines 80 and 518 to amplify the end-goal of early screening (potentially by multi-measurement assays) for improved outcomes.
- The functioning of autophagy is highly complex, and it interacts with several biological processes. This plays a key role in tumor aggressiveness which you have extensively covered in your manuscript. For this reason, include a consideration regarding the processes of apoptosis by including this interesting novel study on the topic (DOI: 10.3390/ijms23073826)
We thank the reviewer for highlighting this helpful reference and have added it to the manuscript (line 512), especially given potential relevance to PCa specificity (as raised by Reviewer 4).
- Check typos
All authors have reviewed the revised manuscript for typos, and we hope that all are now corrected.
Reviewer 4 Report
This research A novel blood proteomic signature for prostate cancer by Muazzam et. al. identified and evaluated thenovel biomarkers associated with prostate cancer patients using SWATH-MS. This is an interesting study. The manuscript is well-organized and clearly written, well-structured, and the results are thoroughly discussed. A very good comparative presentation is shown between healthy and prostate cancer patients. The results reported represent a notable advance in the field of prostate cancer. The authors provide systematic input to the research literature in this field of investigation. However, identified biomarkers have also been well-documented in other cancer and disease studies. The author has not described 12 potential protein markers, is there one that is specifically associated with prostate cancer? The authors also did not indicate whether the study included prostate cancer patients with other disorders/illnesses or who recovered from other disorders/illnesses other than cancer.
Author Response
We thank the reviewers for their time and for the helpful comments. We provide point-by-point responses below.
REVIEWER 4
This research A novel blood proteomic signature for prostate cancer by Muazzam et. al. identified and evaluated the novel biomarkers associated with prostate cancer patients using SWATH-MS. This is an interesting study. The manuscript is well-organized and clearly written, well-structured, and the results are thoroughly discussed. A very good comparative presentation is shown between healthy and prostate cancer patients. The results reported represent a notable advance in the field of prostate cancer. The authors provide systematic input to the research literature in this field of investigation.
However, identified biomarkers have also been well-documented in other cancer and disease studies. The author has not described 12 potential protein markers, is there one that is specifically associated with prostate cancer? The authors also did not indicate whether the study included prostate cancer patients with other disorders/illnesses or who recovered from other disorders/illnesses other than cancer.
The reviewer raises an excellent point. On the specificity to cancer, the data set analysed here did not include participants with other disorders such as benign prostatic hyperplasia (as a specific example of a confounding comorbidity) and so diagnostic specificity with regard to cancer versus this condition was not tested. This is an important limitation and one that we have emphasised further in Discussion to meet the reviewer’s concerns (lines 507-512). We plan to expand the analysis of the SUN biobank data to bring in cohorts with conflicting / confounding comorbidities for our next work, precisely to address this point in a systematic way.
Round 2
Reviewer 3 Report
The manuscript has been deeply improved and, in my opinion, it is worthy for publication